# YC-1 Antagonizes Wnt/β-Catenin Signaling Through the EBP1 p42 Isoform in Hepatocellular Carcinoma

**DOI:** 10.3390/cancers11050661

**Published:** 2019-05-13

**Authors:** Ju-Yun Wu, Yu-Lueng Shih, Shih-Ping Lin, Tsai-Yuan Hsieh, Ya-Wen Lin

**Affiliations:** 1Graduate Institute of Medical Sciences, National Defense Medical Center, Taipei 11490, Taiwan; yun110525@gmail.com (J.-Y.W.); albreb@ms28.hinet.net (Y.-L.S.); 2Division of Gastroenterology, Department of Internal Medicine, Tri-Service General Hospital, National Defense Medical Center, Taipei 11490, Taiwan; tyh1216@ms46.hinet.net; 3Department and Graduate Institute of Microbiology and Immunology, National Defense Medical Center, Taipei 11490, Taiwan; juno1430@gmail.com; 4Graduate Institute of Life Sciences, National Defense Medical Center, Taipei 11490, Taiwan

**Keywords:** hepatocellular carcinoma, YC-1, β-catenin/TCF, ErbB3 binding protein 1 (EBP1), p42 isoform

## Abstract

Novel drugs targeting Wnt signaling are gradually being developed for hepatocellular carcinoma (HCC) treatment. In this study, we used a Wnt-responsive Super-TOPflash (STF) luciferase reporter assay to screen a new compound targeting Wnt signaling. 3-(5′-Hydroxymethyl-2′-furyl)-1-benzylindazole (YC-1) was identified as a small molecule inhibitor of the Wnt/β-catenin pathway. Our coimmunoprecipitation (co-IP) data showed that YC-1 did not affect the β-catenin/TCF interaction. Then, by mass spectrometry, we identified the ErbB3 receptor-binding protein 1 (EBP1) interaction with the β-catenin/TCF complex upon YC-1 treatment. EBP1 encodes two splice isoforms, p42 and p48. We further demonstrated that YC-1 enhances p42 isoform binding to the β-catenin/TCF complex and reduces the transcriptional activity of the complex. The suppression of colony formation by YC-1 was significantly reversed after knockdown of both isoforms (p48 and p42); however, the inhibition of colony formation was maintained when only EBP1 p48 was silenced. Taken together, these results suggest that YC-1 treatment results in a reduction in Wnt-regulated transcription through EBP1 p42 and leads to the inhibition of tumor cell proliferation. These data imply that YC-1 is a drug that antagonizes Wnt/β-catenin signaling in HCC.

## 1. Introduction

Conventional systemic chemotherapy does not increase the survival of patients with hepatocellular carcinoma (HCC). Molecular targeted therapy shows advantages for HCC treatment; however, the effectiveness of targeted therapies is still limited. Sorafenib (Nexavar^®^, Bayer HealthCare Pharmaceuticals, Whippany, NJ, USA), a multiple kinase inhibitor, was the first molecular targeted therapy approved for application in HCC by the U.S. Food and Drug Administration; however, the response rate is actually quite low [1,2]. Therefore, many targeted therapies and combination therapies are still being developed.

HCC carcinogenesis is regulated by a network of various signaling pathways, including Wnt signaling [3,4]. Multiprotein complexes composed of coactivators are required for β-catenin-dependent transcription. To inactivate Wnt signaling, T cell-specific factor/lymphoid enhancer-binding factor (TCF/LEF) interacts with its corepressor, Groucho, and binds to TCF/LEF binding sites within the DNA sequence. Upon Wnt activation, β-catenin is translocated to the nucleus and recruits its coactivators, including Brg1, CBP, Cdc47, Bcl9, and Pygopus. Collectively, these coactivators and β-catenin replace Groucho and interact with TCF/LEF, driving the transcription of Wnt downstream target genes [5,6]. The β-catenin-associated corepressors and coactivators have been increasingly explored. We previously demonstrated that the tumor suppressor SRY-related HMG-box 1 (SOX1) competes with TCF to interact with β-catenin and results in downregulation of the Wnt/β-catenin pathway [7]. Targeting abnormal activation of the Wnt/β-catenin pathway could thus be a therapeutic strategy.

To date, many compounds targeting the β-catenin/TCF complex have been developed and their efficacy evaluated in clinical trials [8]. PKF118-310, PK115-584, and CGPO49090 directly abolish the interaction of β-catenin with TCF/LEF [9]. The combination of sorafenib with β-catenin coactivator inhibitors, such as ICG-001, improves the efficacy of antitumor agents [10]. The mechanism of ICG-001 is based on interference with the interaction between β-catenin and cAMP-responsive element binding (CREB)-binding protein (CBP) [11]. Aberrant Wnt signaling is present in many human cancers, but no drugs have been approved to target the Wnt/β-catenin pathway [8].

In hypoxia, 3-(5′-hydroxymethyl-2′-furyl)-1-benzylindazole (YC-1) is an inhibitor of hypoxia-inducible factor 1-α (HIF-1α) [12]. In addition, YC-1 reduces tumor-associated thrombosis by inhibiting p38/NF-kB activation in hypoxia, but this effect is HIF-1α-independent [13]. In addition, previous studies have revealed that YC-1 is a small compound that can exert antitumor effects in HCC [14,15]. YC-1 is an adjuvant in synergistic therapy by the decreasing of the phosphorylation of signal transducer and activator of transcription 3 (STAT3). YC-1 enhances the induction of apoptosis and the suppression of tumor cell proliferation [15].

Another factor involved in our study is ErbB3 binding protein (EBP1). EBP1, which was identified as an ErbB3 receptor-binding protein, is encoded by the PA2G4 gene. EBP1 is homologous to the 38-kDa murine protein p38-2G4, which is a cell cycle-related protein. EBP1 has two isoforms generated by alternative splicing: p42, the truncated form, and p48, the full-length form. These two isoforms display largely opposing functions in human cancers [16]. While p42 inhibits cell proliferation and promotes differentiation, p48 promotes cell survival through different binding partners and protein modifications in cancer cells [17,18]. Although p48 and p42 play distinct roles, our understanding of the biological significance of these differences is entirely unclear.

In our study, we used a luciferase reporter system to screen 35 compounds from the Library of Pharmacologically Active Compounds (LOPAC). We found that YC-1 did not affect cytosolic and nuclear β-catenin accumulation but enhanced the interaction of ErbB3 receptor-binding protein 1 (EBP1) with the β-catenin/TCF complex. Our data demonstrate that YC-1 suppresses the Wnt/β-catenin signaling pathway and tumor cell proliferation through a novel mechanism.

## 2. Results

### 2.1. YC-1 Inhibits Wnt Signaling and Suppresses Cell Proliferation in HCC

To evaluate the inhibitors of the Wnt/β-catenin pathway, we established a high-throughput screen using a Wnt-responsive Super-TOPflash (STF) luciferase reporter assay in Huh6 cells. The luciferase reporter construct M50-STF, containing 8 copies of the TCF/LEF binding site, was transfected into Huh6 cells to continuously express luciferase. We first screened the potential compounds that significantly reduced the viability of Huh6/M50-STF cells. To screen the potential effective compounds, we treated Huh6/M50-STF cells with the different compounds at a dose of 3 µM for 6 h and determined the efficacy of luciferase activity inhibition. YC-1 was identified as a small molecule inhibitor of the Wnt/β-catenin signaling pathway (Appendix A). Furthermore, we confirmed that 3 µM YC-1 reduced luciferase activity in Huh6/M50-STF cells at 6 and 24 h (Appendix A). To determine whether YC-1 suppressed HCC cell proliferation, the immortalized hepatocyte cell line L-02 and the HCC cell lines HepG2, Huh6 and Hep3B were treated with various concentrations of YC-1 for two durations. The viability of these cells was evaluated by a 3-(4,5-dimethylthiazol-2-yl)-5-(3-carboxymethoxyphenyl)-2-(4-sulfophenyl)-2H-tetrazolium (MTS) assay and was found to be reduced in a YC-1 dose-dependent manner at 24 and 48 h. Only a low level of cytotoxicity to normal L-02 hepatocytes was found after treatment with a high dose of YC-1 (10–100 µM) (Figure 1A). These results showed that HepG2, Huh6, and Hep3B cells exhibit different susceptibilities to the YC-1-mediated reduction in cell viability, with IC_50_ values of 1.54 ± 0.157, 8.8 ± 0.318, and 6.17 ± 0.581 µM, respectively, at the 24-hour time point (Figure 1A and Appendix A). Moreover, treatment with a low dose of YC-1 significantly reduced the viability of HepG2 and Huh6 cells after 72 h. However, prolonged YC-1 treatment did not affect the survival of L-02 cells (Appendix A). Consistent with the cell viability results, the number of colonies formed by HepG2, Huh6, and Hep3B cells was significantly decreased after treatment with the IC_50_ of YC-1 compared with that in the corresponding control cells (Figure 1B). Taken together, these results suggest that YC-1 suppresses HCC cell proliferation.

To further examine the role of YC-1 in the regulation of Wnt signaling, HCC cells were treated with the IC_50_ of YC-1. The effect of YC-1 on Wnt signaling was evaluated by STF luciferase reporter assays. YC-1 significantly decreased the transcriptional activity of TOPflash but not that of the negative control FOPflash in HepG2, Huh6 and Hep3B cells (Figure 2A). Cyclin D1 is the downstream gene of the Wnt signaling pathway [8,19]. Subsequently, we confirmed that YC-1 decreased the expression of cyclin D1 in HepG2, Huh6 and Hep3B cells in a time-dependent manner (Figure 2B). Considering the above results, we suggest that YC-1 effectively reduces the expression of cyclin D1 through the attenuation of Wnt signaling activation, thereby suppressing tumor cell proliferation.

### 2.2. YC-1 Enhances the Recruitment of EBP1 to Interact with the β-Catenin/TCF4 Complex

The formation of the complex containing stabilized nuclear β-catenin and T cell-specific factor 4 (TCF4) triggers the transcription of Wnt target genes and contributes to aberrant activation of Wnt signaling. To investigate the means by which YC-1 suppresses Wnt signaling, we initially investigated the intracellular distribution of β-catenin by immunocytochemical (ICC) analysis. YC-1 did not significantly change the amount of either cytoplasmic or nuclear β-catenin, and this phenomenon was confirmed by western blotting (Appendix A). These results suggested that YC-1 does not affect β-catenin degradation or nuclear β-catenin accumulation. Therefore, we proposed that YC-1 might affect the formation of the β-catenin/TCF4 complex. To avoid capturing the β-catenin degradation complex, we isolated TCF4-binding proteins from HepG2 cells using coimmunoprecipitation (co-IP) and found that YC-1 did not directly disrupt the formation of the β-catenin/TCF4 complex (Appendix A). Next, we used an anti-TCF4 antibody to pulldown proteins in HepG2 cells after YC-1 treatment for comparison with proteins pulled down in control cells. The Coomassie blue staining results showed the presence of unknown proteins in the YC-1-treated cells, and these proteins were analyzed using liquid chromatography-tandem mass spectrometry (LC-MS/MS) (Appendix A). In total, 39 candidate TCF4-binding proteins were identified in YC-1-treated cells. Proliferation-associated protein 2G4 (PA2G4) was identified according to its higher coverage and 5 unique peptides among the TCF4-binding proteins (Appendix A); this strongest potential candidate is also known as ErbB3-binding protein 1 (EBP1). Moreover, by co-IP and western blotting, we confirmed that EBP1 interacted with the β-catenin/TCF4 complex. These data suggested that EBP1 might affect the transcriptional activity of the β-catenin/TCF4 complex. The EBP1 protein has two isoforms, p42 and p48; p48 is the full-length form, and p42 is a truncated form lacking the N-terminus [16]. The protein identification results revealed 5 unique peptides located in the C-terminus and middle regions of EBP1; both p42 and p48 contain these regions (Appendix A). Thus, both isoforms or only the p42 isoform may bind to the β-catenin/TCF4 complex, but p48 does not bind alone.

### 2.3. Knockdown of EBP1 Inhibits the Suppressive Effect of YC-1 in HCC

To determine whether EBP1 is important for the antitumor effect of YC-1, we silenced EBP1 and determined the effect of YC-1 on Wnt signaling and colony formation in HCC cells. We first used shRNA to silence the expression of EBP1 and validated the shRNA knockdown efficiency (Appendix A). The mRNA transcript sequences of the EBP1 isoforms are mostly overlapping; thus, we designed shRNAs targeted to the overlapping region of p42 and p48 mRNA. The protein levels of the two isoforms were simultaneously reduced after shRNA knockdown (Appendix A). For both HepG2 and Huh6 cells, the TOPflash luciferase activity in the YC-1 group was not significantly different from that in the control group when EBP1 was silenced (Figure 3A). Knockdown of EBP1 significantly reversed the pattern of TOPflash luciferase activity in response to 6 h of YC-1 treatment (Figure 3A). Because Wnt signaling was restored in the EBP1-silenced group after YC-1 treatment, we next determined whether EBP1 knockdown influences the suppressive effect of YC-1. EBP1 knockdown affected various cellular functions in the HCC cell lines, including the cell proliferation ability. Therefore, we calculated the difference between the control and YC-1 groups to compare the effects of each shRNA. In the scrambled shRNA groups, the difference between the control and YC-1 treatment was approximately 63.89–99.78%. The difference between the two treatment groups after EBP1 knockdown was approximately 34.85–83.26%. The values of these differences in the EBP1-silenced groups were lower than those in the corresponding scrambled shRNA groups. After the different shRNA groups were treated with YC-1, the colony numbers in the EBP1-silenced groups were partially reversed compared to those in the scrambled shRNA groups (Figure 3B). Similar phenomena were observed in another EBP1-shRNA experiment (Appendix A). These data suggest that the suppressive effects of YC-1 on Wnt signaling and colony formation likely occur through EBP1.

### 2.4. YC-1 Promotes the Interaction of EBP1 p42 with the β-Catenin/TCF4 Complex

EBP1 has two isoforms, p42 and p48, which generally have opposing functions in cells. Gel separation of the p42 and p48 isoforms is difficult due to potential posttranslational modifications. Therefore, we used two antibodies to detect these isoforms of EBP1. The anti-N-EBP1 antibody (labeled N-EBP1 (p48)) is specific for the p48 isoform, whereas the anti-EBP1 antibody can recognize both the p42 and p48 isoforms. The immunoreactive bands detected by the anti-EBP1 antibody are considered the sum of the p42 and p48 bands and labeled EBP1 (p42/p48). EBP1 p42 and p48 were detected in the cytoplasm and nucleus of HCC cells by ICC analysis (Appendix A). We next assessed whether YC-1 affects the amount of nuclear EBP1 isoforms. The western blot results showed that the amount of N-EBP1 (p48) and EBP1 (p42/p48) in the nuclear fraction of HepG2 cells was not significantly different (Appendix A). Therefore, we used co-IP experiments to further determine which EBP1 isoforms mediated the suppressive effect of YC-1. After HCC cells were treated with or without YC-1, the nuclear proteins from HepG2 and Hep3B cells were precipitated by antibodies specific for β-catenin and TCF4. The results showed that both N-EBP1 (p48) and EBP1 (p42/p48) interacted with β-catenin and TCF4. Furthermore, YC-1 enhanced EBP1 (p42/p48) binding to β-catenin and TCF4, but it did not significantly affect N-EBP1 (p48) binding (Figure 4A and Appendix A). Therefore, the increase in the signal corresponding to EBP1 (p42/p48) binding was recognized as EBP1 p42 binding. Taken together, these results prompted us to hypothesize that YC-1 promotes EBP1 p42 binding to the β-catenin/TCF4 complex and subsequently represses the transcriptional activity of the complex in HCC cells.

To further confirm that YC-1 promotes the participation of EBP1 p42 in the β-catenin/TCF complex, we used siRNA (N-si-p48) to deplete the p48 isoform. N-si-p48 greatly reduced the expression of N-EBP1 (p48), and the remaining signal detected by the anti-EBP1 antibody was recognized primarily as associated with the p42 isoform (Appendix A), implying that only p42 was detectable in N-si-p48-silenced cells. To confirm that p42 actually interacted with TCF4 upon YC-1 treatment, we isolated nuclear proteins and performed co-IP experiments. In EBP1 p48-silenced HepG2 cells, the amount of EBP1 (p42/48) was increased in the complexes, as evidenced by immunoprecipitation with anti-β-catenin and anti-TCF4 antibodies (Figure 4B), indicating that the binding of p42 alone is increased. In both the scrambled and N-si-p48 groups of Huh6 and Hep3B cells, the amount of TCF4 was increased in the complexes immunoprecipitated with anti-EBP1 antibodies upon YC-1 treatment (Appendix A). The above results showed that YC-1 enhanced the interaction of the p42 isoform with the β-catenin/TCF4 complex even in the absence of the p48 isoform. Taken together, these findings prompted us to suggest that YC-1 promotes the interaction of EBP1 p42 with the β-catenin/TCF4 complex in HCC cells.

### 2.5. YC-1 Inhibits Colony Formation through EBP1 p42-Mediated Suppression of the Wnt/β-Catenin Signaling Pathway

Next, we aimed to determine whether YC-1 suppresses TCF-dependent transcriptional activity in HCC cells via EBP1 p42. The effect of YC-1 on luciferase activity was determined after depletion of p48 alone in HepG2 and Huh6 cells. We showed that YC-1 attenuated the luciferase activity in the scrambled and N-si-p48 groups, implying that YC-1 suppresses Wnt-regulated transcription mainly through the p42 isoform (Figure 5A). Moreover, YC-1 inhibited colony formation in both the scrambled and N-si-p48 groups. Interestingly, the colony numbers were markedly decreased in N-si-p48 cells. Colony formation was significantly reduced in p48-silenced cells treated with YC-1 compared to that in p48-silenced cells not treated with YC-1 (Figure 5B). Taken together, these results suggest that YC-1 reduces Wnt-regulated transcription through EBP1 p42 and inhibits tumor cell proliferation.

## 3. Discussion

Various compounds to treat HCC have been synthesized, and some are approved for marketing by the U.S. Food and Drug Administration. To identify new drugs, we used a high-throughput reporter system to discover a compound targeting Wnt signaling. Currently, no reports demonstrate that YC-1 can repress the Wnt/β-catenin pathway. Interestingly, YC-1 was identified as an effective antagonist that rapidly repressed luciferase activity. Our data demonstrate that YC-1 suppresses Wnt/β-catenin transcriptional activity and tumor cell proliferation by increasing the binding of the p42 isoform to the β-catenin/TCF complex.

YC-1 is an activator of soluble guanylyl cyclase (sGC) that increases the level of intracellular cyclic guanosine monophosphate (cGMP), resulting in inhibition of blood coagulation [20]. Recent studies have reported that the mechanism underlying the antitumor effect of YC-1 is HIF-1α inhibition in hypoxia as well as STAT3 activation in normoxia [12,14,15]. Interestingly, our data showed that YC-1 did not affect the expression of HIF-1α (Appendix A), suggesting that other mechanisms are involved. Additionally, HIF-1α expression was not induced in the scrambled shRNA-transfected group upon YC-1 treatment; this result ensures that the signaling pathways were not affected by interference from other factors (Appendix A). Here, we demonstrated that YC-1 suppresses tumor cell proliferation through reducing the expression of Wnt target genes such as cyclin D1. Importantly, no serious toxicity was observed in immortalized hepatocyte L-02 cells during treatment with YC-1, similar to the observation in mice [21,22]. These data provide evidence supporting YC-1 as a potential drug antagonizing Wnt/β-catenin signaling in HCC.

First, we speculated that YC1 exerted an antitumor effect by reducing the accumulation of nuclear β-catenin or suppressing the dissociation of the β-catenin/TCF complex. In our study, YC-1 could not promote β-catenin translocation, and it interfered with the interaction between β-catenin and TCF/LEF. Therefore, we hypothesized that YC-1 might recruit repressors to prevent β-catenin/TCF complex assembly. We used co-IP and LC-MS/MS to identify EBP1 as an interacting partner of the β-catenin/TCF complex. YC-1 enhanced the recruitment of EBP1, and these mediators collaborated to suppress Wnt signaling.

Endogenous EBP1 is found in human embryonic stem cells (hESCs), human induced pluripotent stem cells (IPSCs) and human embryonic carcinoma cells (hECCs) [23]. EBP1 is also distinctly expressed in various kinds of cancers, such as bladder cancer [24], glioblastoma [25], non-small cell lung carcinoma (NSCLC) [26], cervical cancer [27], acute myeloid leukemia (AML) [28], salivary adenoid cystic carcinoma [29], colon cancer [30] and prostate cancer [31,32]. EBP1 encodes two isoforms that regulate opposing cellular functions. EBP1 p48 promotes cell proliferation and tumorigenesis in glioblastoma, AML, and colon cancer, acting as an oncogenic regulator. By contrast, EBP1 p42, acting as a tumor suppressor, inhibits cell proliferation, invasion and tumorigenesis in NSCLC. A previous study demonstrated that downregulation of EBP1 is correlated with poor prognosis and that depletion of EBP1 enhanced cell proliferation in HCC [33]. Therefore, we hypothesized that the p42 isoform might inhibit cell proliferation in HCC. Then, we determined the isoform of EBP1 that regulates the cellular response induced by YC-1. To avoid capturing cytoplasmic EBP1, we used the nuclear extract for co-IP experiments. Anti-EBP1 antibodies were used to detect p42 because no antibodies specific for p42 have been produced. Owing to the difficulties in detecting p42, the increase in the EBP1 (p42/48) signal detected by the anti-EBP1 antibody was proposed to be associated with p42 binding (Figure 4, Appendix A). YC-1 treatment did not change the interaction with p48. However, EBP1 (p42/p48) binding was increased, indicating that p42 was increasingly assembled into a complex with β-catenin and TCF4. Furthermore, we silenced only p48 and confirmed that YC-1 increased the binding capability of p42.

Surprisingly, the luciferase reporter activity and colony numbers were significantly decreased only in p48-silenced cells, possibly because p48 functions as an oncogene. EBP1 p48 physically interacts with cyclin-dependent kinase 2 (CDK2) [34], protein kinase B (PKB/Akt) [35], Akt/murine double minute 2 (MDM2) [36] and F-box and WD repeat domain-containing 7 (FBXW7) [30]. EBP1 p48 contributes to p53 degradation, cell proliferation and the apoptosis suppression and is associated with Akt signaling. Disruption of the interaction between Akt and p48 might facilitate the stabilization of the p53 protein. Fbxw7 is a tumor suppressor in HCC whose activation is increased by p53, thereby inhibiting tumorigenesis [37]. Thus, the interaction of p48 with Akt might affect the induction of apoptosis and the suppression of cell proliferation in HCC. In addition, evidence indicates that protein kinase C (PKC) can phosphorylate EBP1 at the Ser363 site in the C-terminus of EBP1 [38] and that Ser363 phosphorylation results in the inhibition of tumor cell proliferation [39]. Moreover, YC-1 increases the activity of PKC [40]. In the future, we are interested in determining whether the interaction of p42 with the β-catenin/TCF complex is regulated by PKC in response to YC-1.

## 4. Materials and Methods

### 4.1. Cell Culture and Reagents

The HepG2, Huh6 and Hep3B human HCC cell lines and the L-02 immortalized hepatocyte cell line were used in this study. HepG2 and Hep3B cells were purchased from the American Type Culture Collection (ATCC, Rockville, MD, USA). HuH6 cells were obtained from Professor K.H. Lin (Chuang-Gung University, Taiwan). The L-02 immortalized hepatocyte cell line was obtained from Professor Alfred S.L. Cheng (Institute of Digestive Disease and Li Ka Shing Institute of Health Sciences, The Chinese University of Hong Kong, Hong Kong, China). These cells were cultured in Dulbecco’s modified Eagle’s medium (GIBCO, Gaithersburg, MD, USA) supplemented with 10% fetal bovine serum (GIBCO) and 0.1% penicillin-streptomycin (GIBCO) at 37 °C under 5% CO_2_.

The LOPAC1280 library and YC-1 were purchased (Sigma-Aldrich, St. Louis, MO, USA). YC-1 was prepared in dimethyl sulfoxide (DMSO) (Sigma-Aldrich) at a concentration of 30 mM. Cells were treated with the IC_50_ of YC-1 in complete medium. Cells treated with DMSO served as the control groups.

### 4.2. The Delivery of Short Hairpin RNA (shRNA) or Dicer-Substrate siRNA (DsiRNA)

The individual oligo sequences of the EBP1 shRNAs are listed in the Table 1. The constructs were obtained from the National RNAi Core of Taiwan. N-si-p48 targeting EBP1 p48 alone was synthesized (Integrated DNA Technologies, Coralville, IA, USA). The oligo sequences of N-si-p48 are 5′-CAGCGAUAGUUUGCUCCUGUUGCUCGU-3′ (antisense) and 5′-GAGC AACAGGAGCAAACUAUCGCTG-3′ (sense). N-si-p48 was designed to target the region upstream of the nucleotide skipping region; thus, it silenced only EBP1 p48.

EBP1 shRNA expression plasmids or N-si-p48 DsiRNAs were transfected into cells using TransIT-X2 transfection reagent (Mirus Bio LLC, Madison, WI, USA). Cells were transfected with 2 μg of the EBP1 shRNA expression plasmid or 25 nM synthetic N-si-p48. The silencing efficiency was assessed by western blotting.

### 4.3. TCF/LEF Luciferase Assay

The pGL4.21 [luc2P/Puro] vector was purchased from Promega (Madison, WI, USA). The M50 Super 8× TOPflash and M51 Super 8× FOPFlash vectors were obtained from the Randall T. Moon laboratory. The sequences of the TCF/LEF binding sites in the M50 Super 8× TOPflash and M51 Super 8× FOPFlash vectors were cloned into the pGL4.21 vector. The pGL4.21-TOPflash and pGL4.21-FOPflash vectors were generated and used in this study.

To establish the Huh6/M50 stable cell lines, Huh6 cells were transfected with the pGL4.21-TOPflash vector and incubated for 2 days. The transfected cells were selected with 0.5 mg/mL puromycin (Sigma-Aldrich) until colonies formed. A ONE-Glo Luciferase Assay System (Promega, Madison, WI, USA) was used to select the cell colonies with high luciferase activity.

To assess the effect of shRNA and DsiRNA on Wnt-regulated transcription, cells were cotransfected with the pGL4.21 vector containing the TOPflash or FOPflash sequence and the indicated shRNA or DsiRNA. The experiments required transfection of the pRL-TK Renilla luciferase vector as the control for the transfection efficiency. The firefly and Renilla luciferase activities were measured by a Dual-Glo Luciferase Assay System (Promega). The relative activity of TOPflash or FOPflash was normalized to the Renilla luciferase activity and indicated the fold change in Wnt/β-catenin pathway activation.

### 4.4. Cell Viability Assay

Cells were seeded at a density of 1 × 10^4^ cells per well in a 96-well plate and cultured overnight. These cells were treated with DMSO or YC-1 (Sigma-Aldrich) at the indicated concentrations for 24 and 48 h. The cell viability was measured by an MTS assay (Sigma-Aldrich). All experiments were performed in triplicate.

### 4.5. Clonogenic Assay

Cells at the optimal density of 5 × 10^4^ to 1 × 10^5^ cells per well were seeded in 6-well plates and cultured overnight. These cells were treated with DMSO or YC-1 for 24 h, and the medium was then replaced with fresh complete medium. The treated cells were maintained until colonies formed. Colonies were stained with crystal violet (Sigma-Aldrich) and counted by ImageJ software (National Institute of Mental Health, Bethesda, MD, USA).

### 4.6. Immunocytochemical (ICC) Staining

Cells were seeded on 12-mm glass coverslips and cultured overnight. After treatment with DMSO or YC-1, cells were washed with phosphate-buffered saline (PBS) and fixed in 4% formaldehyde solution (Sigma-Aldrich) at room temperature for 20 min. Fixed cells were washed and permeabilized with 0.5% Triton X-100 in PBS for 15 min at room temperature. Permeabilized cells were washed and blocked with 1% BSA in PBS overnight at 4 °C. Primary antibodies were added at a 1:200 dilution and incubated overnight at 4 °C Cells probed with the primary antibodies were washed with PBS three times. Secondary antibodies were diluted 1:300 and incubated at room temperature for 1 h. Cells were washed and stained with Hoechst 33342 (bisbenzimide H33342 trihydrochloride) (Sigma-Aldrich) to visualize nuclei. Stained cells on coverslips were mounted to slides using Fluoromount™ Aqueous Mounting Medium (Sigma-Aldrich). Fluorescence images were acquired with an LSM 880 confocal laser scanning microscope with Airyscan (Carl Zeiss Inc., Germany). The primary antibodies used were as follows: mouse anti-β-catenin (BD Biosciences, CA, USA), mouse anti-N-EBP1 (C11) (Santa Cruz, CA, USA), rabbit anti-EBP1, and rabbit anti-histone H3 (GeneTex, Irvine, CA, USA). FITC-conjugated anti-mouse IgG (Bethyl Laboratories, TX, USA) and DyLight 594-labeled anti-rabbit IgG (GeneTex) secondary antibodies were used as appropriate.

### 4.7. Nuclear Extraction

Nuclear extracts were prepared using a Nuclear Extraction Kit 2900 (Millipore Corporation, Billerica, MA, USA) according to the manufacturer’s protocol. After treatment with DMSO or YC-1, cells were washed with PBS and collected. Cytoplasmic proteins were isolated in cytoplasmic lysis buffer containing 0.5 mM DTT and protease inhibitor. Cell pellets were washed again with cytoplasmic lysis buffer. Nuclear proteins were isolated in nuclear lysis buffer containing 0.5 mM DTT and protease inhibitor. The concentration of nuclear protein was measured with a Pierce BCA Protein Assay Kit (Thermo, Waltham, MA, USA).

### 4.8. Coimmunoprecipitation Assays and Mass Spectrometry

IP assays were conducted as described previously [7]. A 150-µg sample of nuclear protein was suspended in IP buffer containing 10 mM Tris-HCl (pH 7.6), 145 mM NaCl, 1 mM EDTA, 1 mM EGTA, 0.5% Triton X-100 (Sigma-Aldrich) and protease inhibitors (Millipore). The nuclear protein was incubated with primary antibodies overnight at 4 °C on a rotator. Before the pulldown of nuclear complexes, protein A/G Sepharose beads (Abcam, Cambridge, UK) were washed with IP buffer. For blocking, the beads were added to IP buffer supplemented with 1 mg/mL BSA (Sigma-Aldrich). After 1 h, the BSA-blocked beads were added to the complex-conjugated antibodies. The beads and antibodies were gently mixed for 1 h at 4 °C and were then centrifuged at 5000× *g* for 5 min at 4 °C. The supernatant was discarded. The pelleted beads were washed three times with IP buffer, resuspended in Pierce IP lysis buffer (Thermo) and boiled at 95 °C for 10 min. The beads were then cooled on ice for 20 min and centrifuged at 8000× *g* for 5 min at 4 °C. The eluted proteins were then collected and analyzed by LC-MS/MS and western blotting.

The eluted proteins were submitted to the Proteomics Core (National Taiwan University Center of Genomic and Precision Medicine, Taipei, Taiwan) for LC-MS/MS analysis on an LTQ-Orbitrap Velos spectrometer. Peptide fragments were identified by an automated search (Mascot software, Matrix Science, Boston, MA, USA) against the NCBI protein database.

### 4.9. Western Blotting

Western blot assays were conducted as described previously [7]. Protein samples were separated by sodium dodecyl sulfate-polyacrylamide gel electrophoresis (SDS-PAGE) on 8% or 10% gels (Thermo) and were then transferred to polyvinylidene difluoride (PVDF) membranes (Millipore Corporation, Billerica, MA, USA). PVDF membranes were blocked in 1 mg/mL BSA (Sigma-Aldrich) at room temperature for 1 h. Primary antibodies were diluted 1:2000 and incubated overnight at 4 °C. Secondary antibodies diluted 1:5000 were added and incubated at room temperature for 1 h. The band intensities corresponding to the protein samples were measured using Immobilon Western Chemiluminescent HRP Substrate (Millipore). The primary antibodies used were as follows: mouse anti-β-catenin, mouse anti-lamin A/C (BD Biosciences, CA, USA), mouse anti-TCF4 clone 6H5-3 (Millipore), mouse anti-N-EBP1 (C11) (Santa Cruz, CA, USA), rabbit anti-EBP1, rabbit anti-histone H3, rabbit anti-α-tubulin, rabbit anti-HIF-1α and rabbit anti-β-actin (GeneTex, CA, USA). Horseradish peroxidase-conjugated rabbit anti-mouse or goat anti-rabbit secondary antibodies (Santa Cruz Biotechnology) were used as appropriate.

### 4.10. Statistical Analysis

Significant differences were analyzed by two-tailed unpaired Student’s *t*-test using GraphPad Prism (GraphPad Software Inc., La Jolla, San Jose, CA, USA). The data are expressed as the mean ± SEM. *p* < 0.05 was considered significant.

## 5. Conclusions

In summary, we identified a novel mechanism by which YC-1 suppresses Wnt signaling in HCC. YC-1 effectively suppresses tumor proliferation through repressing the transcription of Wnt-regulated genes such as cyclin D1. Subsequently, we demonstrated the importance of EBP1 participation in the β-catenin/TCF interaction. Upon YC-1 treatment, p42 increasingly interacted with the β-catenin/TCF complex. The interaction with p42 results in suppression of tumor cell proliferation. These data suggest that YC-1 can effectively suppress the Wnt/β-catenin pathway via EBP1 p42 in HCC therapy (Figure 6).

## Figures and Tables

**Figure 1 cancers-11-00661-f001:**
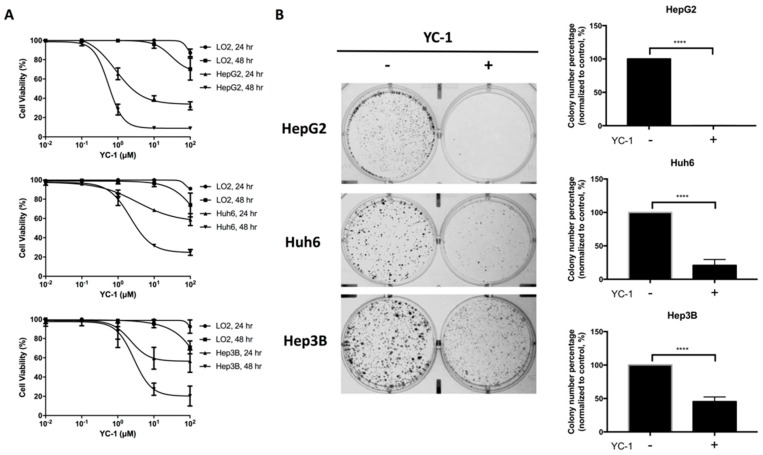
YC-1 suppressed hepatocellular carcinoma (HCC) cell proliferation in a dose- and time-dependent manner. HCC cells were exposed to serial dilutions of YC-1 ranging from 0.01 to 100 µM. After 24 and 48 h, the viability of HepG2, Huh6, and Hep3B cells was measured by an MTS assay (**A**). The effect of YC-1 on the cell colony formation was assessed by a clonogenic assay. HCC cells were treated with dimethyl sulfoxide (DMSO) or with the IC_50_ of YC-1 for 24 h, and the medium was then replaced with complete medium. After 10 days, colonies were fixed and stained (**B**). The error bars indicate the standard error of mean (SEMs) of data obtained in at least three independent experiments. **** *p* < 0.0001.

**Figure 2 cancers-11-00661-f002:**
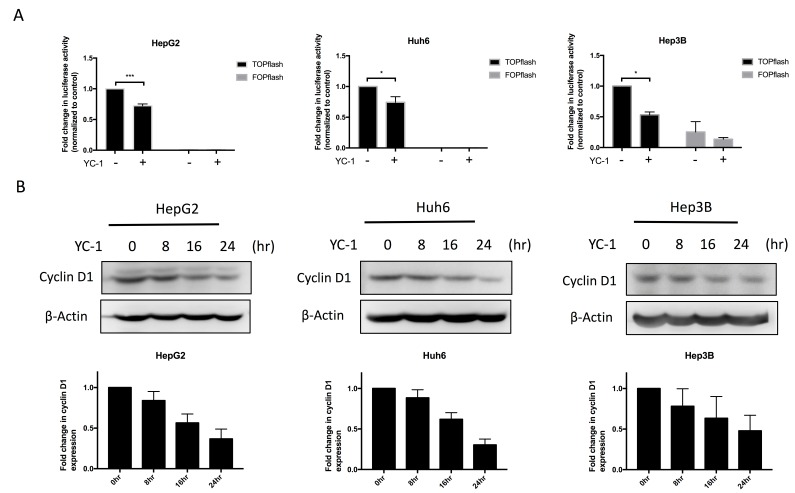
YC-1 inhibited Wnt signaling and cyclin D1 expression. The TOPflash reporter containing wild-type TCF/LEF binding sites produced a high level of transcriptional activity in HCC cells. The FOPflash reporter containing the mutated TCF/LEF binding sites was used as the negative control. The luciferase activity of TOPflash and FOPflash was analyzed after 6 h of treatment with the IC_50_ of YC-1 (**A**). All HCC cell lines were exposed to the IC_50_ of YC-1 for the indicated durations. The expression of cyclin D1 was analyzed by western blotting (**B**). * *p* < 0.05, *** *p* < 0.001.

**Figure 3 cancers-11-00661-f003:**
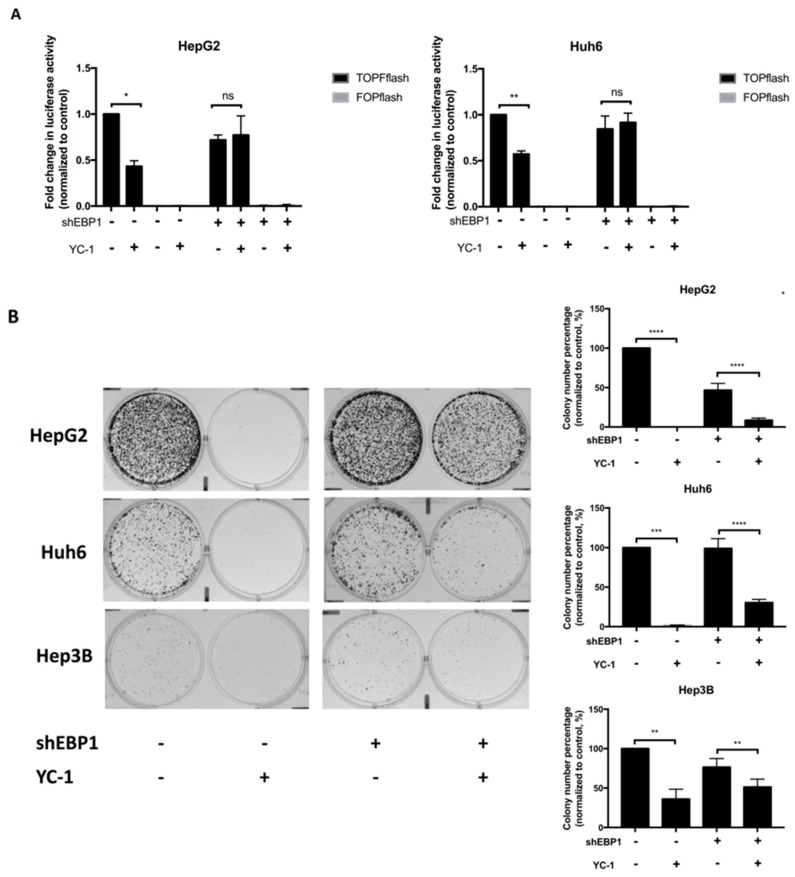
Knockdown of EBP1 inhibited the suppressive effect of YC-1 on HCC cells. HCC cells were transfected with scrambled and EBP1-specific shRNA #2 for 24 h and recovered for 48 h. Scrambled shRNA-transfected and EBP1-silenced cells were treated with the IC_50_ of YC-1 for 6 h. Luciferase activity was measured by STF reporter assays (**A**). The indicated cells were seeded in 6-well plates at 5 × 10^4^ cells per well and incubated for 48 h. After the HCC cells were treated with the IC_50_ of YC-1 for 24 h, the medium was replaced. The colonies formed from the indicated cells were observed and counted (**B**). * *p* < 0.05, ** *p* < 0.01, *** *p* < 0.001, **** *p* < 0.0001, ns: *p* > 0.05.

**Figure 4 cancers-11-00661-f004:**
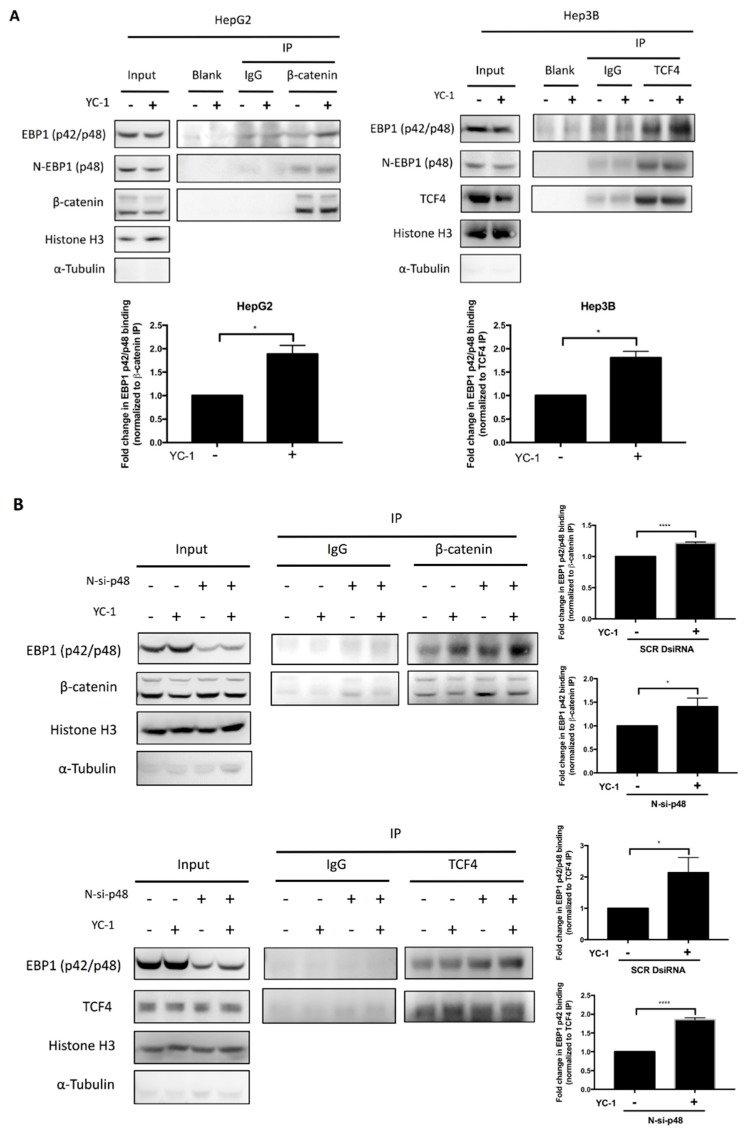
YC-1 enhanced the interaction of EBP1 p42 with the β-catenin/TCF complex in HCC cells. HepG2 and Hep3B cells were treated with the IC_50_ of YC-1 for 6 h. The nuclear complexes containing the EBP1 isoforms were precipitated by anti-β-catenin and anti-TCF4 antibodies. The anti-N-EBP1 antibody is specific for p48, whereas the anti-EBP1 antibody recognizes both the p42 and p48 isoforms. The immunoreactive bands detected by the anti-EBP1 antibody are considered the sum of the p42 and p48 bands. The levels of EBP1 isoform binding to β-catenin and TCF4 were measured by western blotting (**A**). N-si-p48 DsiRNA targeting p48 was transfected into HepG2 cells for 24 h. Under conditions of p48 depletion alone, the interaction of EBP1 (p42/p48) with the β-catenin/TCF complex was confirmed in HepG2 cells treated with or without YC-1 (1.54 μM). The binding capability of nuclear p42 was detected by co-IP and western blotting (**B**). * *p*< 0.05, **** *p* < 0.0001.

**Figure 5 cancers-11-00661-f005:**
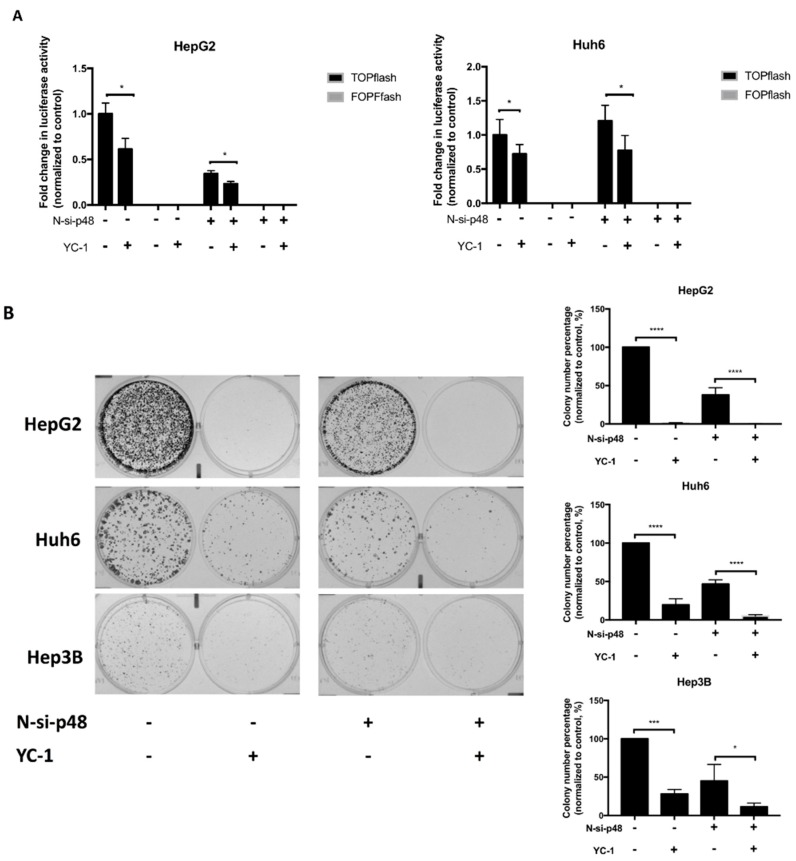
YC-1 inhibited colony formation through EBP1 p42, suppressing the Wnt/β-catenin pathway in HCC cells. Scrambled and N-si-p48 DsiRNA were transfected into HepG2 and Huh6 cells along with either the TOPflash or FOPflash vector following the indicated treatments. These transfected cells were treated with the IC_50_ of YC-1 for 6 h, and luciferase activity was measured (**A**). Cells transfected with different DsiRNAs were seeded in 6-well plates at 5 × 10^4^ cells per well and incubated for 48 h. These cells were treated with the IC_50_ of YC-1 for 24 h, and the medium was then replaced with fresh medium. After one to two weeks, colonies were stained and counted. The error bars indicate the SEMs of data obtained in at least three independent experiments (**B**). * *p* < 0.05, *** *p* < 0.001, **** *p* < 0.0001.

**Figure 6 cancers-11-00661-f006:**
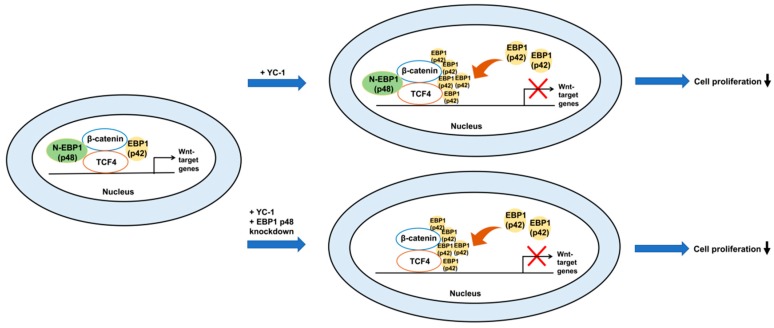
Schematic model of YC-1 involvement in the regulation of Wnt/β-catenin signaling and tumor cell proliferation.

**Table 1 cancers-11-00661-t001:** Oligosequences of the shRNAs.

Name	Sequence
Scrambled shRNA	CCGGTGTTCGCATTATCCGAACCATCTCGAGATGGTTCGGATAATGCGAACATTTT
shEBP1#1	CCGGCCACCAGCATTTCGGTAAATACTCGAGTATTTACCGAAATGCTGGTGGTTTTTG
shEBP1#2	CCGGCGCTAATGTAGCTCACACTTTCTCGAGAAAGTGTGAGCTACATTAGCGTTTTTG
shEBP1#3	CCGGCCTGGTCGTGACCAAGTATAACTCGAGTTATACTTGGTCACGACCAGGTTTTTG
shEBP1#4	CCGGAGGACAGAGAACCACTATTTACTCGAGTAAATAGTGGTTCTCTGTCCTTTTTTG

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
