# Peer review of "YC-1 Antagonizes Wnt/β-Catenin Signaling Through the EBP1 p42 Isoform in Hepatocellular Carcinoma"

_cancers, 2019, doi:10.3390/cancers11050661_

Round 1

Reviewer 1 Report

Wnt/β-Catenin signaling pathway is critical for development of hepatocellular carcinoma (HCC).  Targeting Wnt/β-Catenin signaling pathway may effectively block hepatocellular carcinoma. The authors found that a small molecule inhibitor, YC-1, can antagonize Wnt/β-Catenin signaling pathway leading to an inhibition of growth of three HCC cell lines in vitro. Furthermore, the authors demonstrated that YC-1 inhibits  Wnt/β-Catenin signaling pathway through the EBP1 p42 isoform rather than p48 isoform. Also, the authors have shown that the function of YC-1 in blocking cancer cells proliferation is independent of HIF-1a signaling pathway. The authors provided a new mechanism for YC-1 in targeting HCC. However, there is no justification of YC-1 was selected from the 1280 compound library. In addition, the data can not support the conclusion that YC-1 suppresses Wnt activity through the binding of p42 isoform to the β-catenin/TCF complex.  Other data for HIF1a signaling pathway were also not convincing. My concerns are below.

1.       In Line 72, the authors used luciferase reporter system to screen 1280 compounds from the LOPAC library (Sigma). They claimed that YC-1 was identified as a small molecule inhibitor of  the Wnt/β-catenin signaling pathway in Supplementary Figure 1. However, there is no data showing that YC-1 is the most effective in reducing luciferase activity among the above compounds.

2.       In lines 98 - 99, why cyclin D1 was selected as the most important target of Wnt signaling pathway? What are other downstream targets of Wnt signaling pathway besides cyclin D1?

3.       In line 125, the authors did not validate the function of PA2G4 in HCC proliferation and a correlation between PA2G4 with ErbB3-binding protein 1 (EBP1).

4.       in Figure 3B, right panel, the comparison did not make sense. It should be compared between the 2nd column (shEBP1 negative and YC-1 positive) and the 4th column (shEBP1 positive and YC-1 positive). In addition, at least two shRNAs should be used for functional assays.

5.       In line 172, what concentrations of YC-1 promoted EBP1 p42 binding to β-catenin /TCF4 complex?

6.       In line 191 and Figure 4B, if p48 isoform did not affect the binding to the β-catenin and TCF4, why knockdown of p48 combined with YC-1 could increase the amounts of β-catenin and TCF4 compared with YC-1 treatment alone?

7.       In lines 201 – 207 and Figure 5A, the conclusion did not make sense. First, knockdown of p48 isoform enhanced decreases of colony formation compared to single treatment with YC-1. Second, the authors did not use siRNA against p42 isoform. How did they draw a conclusion that the function of YC-1 in suppression of Wnt signaling pathway is mainly through the p42 isoform?

8.     In Supplementary Figure 8, YC-1 is supposed to inhibit HIF-1a expression in hypoxia conditions referred to the literatures (12, 13, 18) . However, the data that the cells were treated with scrambled control at the same time points were not shown. It is not convincing to draw this conclusion that YC-1 did not affect the expression of HIF-1a.

Reviewer 2 Report

                In this paper, Wu et al. demonstrate that YC-1, a compound identified as a small molecule inhibitor of the Wnt-β catenin (Wnt) pathway, recruits the p42 isoform of Ebp1 to the Wnt complex. The recruitment of p42 Ebp1 leads to repression of Wnt signaling and growth inhibition. This paper is of interest as it demonstrates a new function for Ebp1 and also sheds light on a mechanism of Wnt signaling regulation.  However, revision is needed in certain aspects of experimental design and data presentation.

Introduction The authors should include a brief summary of  the function of Ebp1 and its two isoforms.

Results

The authors bring up the valid point of the technical difficulties in identifying the two isoforms of Ebp1 using the rabbit antibody.  However, different types of gel percentages can reveal the two forms (Ahn et al ). It would be important to show the relative levels of the two isoforms in HCC cells before and after treatment with YC-1. In general, the p42 isoform is reduced in different types of cancers.  Thus, it is difficult to appreciate what the physiological function of the p42 isoform might be in HCC.

Figure 1A.  Viability assays should be performed at longer time points.

Fig. 1B The authors have used a relatively high concentration of cells (3 x 105) for the colony assays.  Quantitation of the colony assays is difficult at this concentration.  These assays should be rerun here and in all subsequent figures at a lower concentration of cells.  In addition, quantitative results should be included for the colony assays.

Fig. 2 The regulation of Cyclin D1 expression by Ebp1 has already been reported and should be cited (Zhang et al.  Nuc. Acids. Res 31:2168).

Fig. 3 The restoration in colony growth after YC-1 treatment after knock out of Ebp1 appears to vary from cell line to cell line.  Do the authors have any explanation of this?

Supplementary data S5.  Why was the N terminal antibody used to measure knock out of Ebp1?  This would not measure changes in p42 expression levels and p42 is supposed to be the mediator of Ebp1’s effects in this system.

Fig. 4  The authors deduce , using both the N terminal and rabbit Ebp1 antibody, that YC-1 promotes the binding of p42 Ebp1 to the β- catenin complex.  This is a fair assumption. However, have the authors ever used overexpression of a tagged p42 Ebp1 to demonstrate binding to the  β-catenin complex? Successful completion of this experiment would strengthen their arguments.

Discussion Do the authors have any hypothesis explaining how YC-1 might trigger the association of Ebp1 with the Wnt complex?

The paper is generally well-written, but should be edited for minor errors in English style and usage. 

Round 2

Reviewer 1 Report

I am satisfied with the revision. 

Reviewer 2 Report

The authors have responded appropriately to the comments.  The paper is now suitable for publication.